# Borderline Personality Symptoms: What Not to Be Overlooked When Approaching Suicidal Ideation among University Students

**DOI:** 10.3390/healthcare9101399

**Published:** 2021-10-19

**Authors:** Nahathai Wongpakaran, Awirut Oon-Arom, Nuntaporn Karawekpanyawong, Trustsavin Lohanan, Thanakorn Leesawat, Tinakon Wongpakaran

**Affiliations:** 1Department of Psychiatry, Faculty of Medicine, Chiang Mai University, Chiang Mai 50200, Thailand; nahathai.wongpakaran@cmu.ac.th (N.W.); awirut.oonarom@elearning.cmu.ac.th (A.O.-A.); nuntaporn.karawek@cmu.ac.th (N.K.); 2Faculty of Medicine, Chiang Mai University, Chiang Mai 50200, Thailand; trustsavin_lohanan@cmu.ac.th (T.L.); thanakorn_l@cmu.ac.th (T.L.)

**Keywords:** borderline personality disorder, screening, validation, instrument, undergraduates

## Abstract

Suicidal ideation is a serious condition antecedent to suicidal attempts and is highly related not only to depression but also other psychosocial factors. This study aimed to examine the predictive effects of these potential factors for suicidal ideation among young adult university students. A cross-sectional survey was conducted on a sample of university students in Thailand. An online questionnaire employed the perceived stress scale-10 (PSS-10), the patient health questionnaire-8 (PHQ-8), the multidimensional scale of perceived social support (MSPSS), and a screening instrument for borderline personality disorder. An ordinal regression analysis was applied to determine the predictive effects of the independent variables. Of 336 students, the mean age was 20.26 ± 1.3 years, 80.4% of whom were female; 14.3% had suicidal ideation. The significant predictors of suicidal thoughts were perceived stress (AOR 1.11, 95% CI 1.01 to 1.22); depressive symptoms (AOR 1.16, 95% CI 1.05 to 1.22); borderline personality symptoms (AOR 1.19, 95% CI 1.01 to 1.40); and perceived social support (AOR 0.97, 95% CI 0.94 to 1.00). Not only did depressive symptoms contribute to suicidal ideation but they also constituted important variables. Therefore, they should be included in intervention plans to prevent suicidality among university students.

## 1. Introduction

In the transition from adolescence to adulthood, university students often are required to adapt to new social roles, to deal with their own finances, and to strive for academic achievement [1]. Some may undergo a mental health hurdle such as stress, amotivation, interpersonal difficulty, or depression [2,3,4]. The mental health problems of university students have strikingly increased in the past decade [5,6], especially among medical students. A systematic review has reported that the prevalence of depression among university students ranged from 10 to 85% [7].

One serious element of psychological distress is the suicide attempt, for which empirically, the most potent predictor is suicidal ideation [8]. A study in the US found that 24% of undergraduate subjects had suicidal ideation and 9% had attempted suicide [5], compared with a 9% prevalence of suicidal ideation, and 2.7% of suicide attempts among adults worldwide [9]. Suicidal ideation is omnipresent among university students across culture and religion. A survey among Muslim university students showed that 22% of the participants reported suicidal ideation and 8.6% reported attempting suicide [10]. In some countries, studies showed that the prevalence of suicidal ideation and attempt were 23.7 and 3.9%, respectively [11]. In Thailand, the prevalence of suicidal ideation in the past 12 months was 8.8% (9.9% among males and 7.7% among females) [12].

In addition to sociodemographic variables, e.g., being female [6], many factors are associated with suicidal ideation, especially psychosocial variables that typically accompany this developmental period [13]. Studies have verified that university students experience high levels of stress [2,14], which are strongly associated with a greater likelihood of suicidality [5]. Sources of stress included academic, family-related, and relationship-related causes. One half of students reported the presence of academic stress as an important life stressor leading to suicidality [15].

Depression may be the most common variable related to both suicidalities, either suicidal ideation or suicide attempt [6,16,17,18]. Related studies have reported a wide range of the prevalence of depression among university students, from 10 to 85% [7]. Similar to studies in Western countries, depression constituted a significant predictor of high suicide attempt risk as indicated in Asian countries [19,20]. The incidence strikingly increased during the COVID-19 outbreak. A study showed that during lockdown, major depression and suicidal attempts were present among 12.43 with 13.46% of subjects experiencing severe distress [21,22].

In addition to stress and depression, perceived social support, a positive factor, was found to be negatively associated with suicidal ideation. Many studies have revealed that the odds of feeling poor social support involved two to five times increased odds of suicide ideation [6,11]. Individuals with feelings of poor social support had two to five times the odds of suicidal ideation than those who had well-perceived social support. This finding was consistent between Western and Eastern cultures [19,23,24,25,26].

One important factor, related to suicidal ideation, is childhood history of abuse and maltreatment [16,27,28,29,30]. This factor is fundamentally associated with borderline personality disorder (BPD) in older age [31]. The core features of borderline personality include unstable interpersonal relationships, poor self-image, and affects, and marked impulsivity [32], which usually become apparent during adolescence and young adulthood [33,34].

A review of 43 studies discovered that the prevalence of BPD among university students was 0.5 to 32.1%, with an unadjusted lifetime prevalence of 9.7% [35]. One of the most singular aspects of BPD among college students was chronic suicidal ideation due to being associated with poor self-concept and identity [36,37,38], and was predictive of a forthcoming suicide attempt [39,40].

In addition to suicidality, BPD is closely related to high levels of perceived stress, depression, and a low level of perceived social support [40,41,42,43,44].

As the rate of increasing suicidal ideation and attempts has become a major concern among university students, many studies have focused on suicidal ideation, most emphasizing depression, but insufficiently on psychosocial issues, i.e., perceived stress and perceived social support that is deemed modifiable [45,46]. Many studies have indirectly investigated personality factors, including several in Thailand, but without exploring borderline personality directly [12,47]. Due to adapting to a transitional stage of life and encountering environmental challenges, some university students with certain personality types may experience tremendous distress; therefore, including the personality factor would be important when studying suicidality among university students. More importantly, to our knowledge, the modifiable positive and negative psychosocial factors have not been analyzed concurrently with borderline personality among university students. Therefore, this study aimed to explore these potential predictors related to suicidal ideation. We hypothesized that all of these psychosocial variables may constitute significant predictors for suicidal ideation among Thai university students.

## 2. Materials and Methods

This study employed a cross-sectional online survey of 336 undergraduate students in Thailand in late 2019 before the COVID-19 pandemic in Thailand. It comprised a cross-sectional online survey. The study was approved by the Ethics Committee of the Faculty of Medicine at Chiang Mai University, Thailand.

Participants

Undergraduate students throughout Thailand were invited to participate in the study. The inclusion criteria included age 18 to 25 years, studying at an undergraduate level, fluent in writing and speaking Thai, and able to access the Internet. The exclusion criteria included being diagnosed with schizophrenia, bipolar disorder, drug or alcohol use disorder, and being intoxicated with alcohol within 24 h before participating in the study. After finalizing a written informed consent form, the participants were asked to complete questionnaires on the Internet via personal computer, laptop, smartphone, or tablet. The amount of 100 THB (3 USD) was given to each participant for compensation after they had submitted their answers.

The target sample size of participants was estimated based on related studies among university students. As reviewed, the range of the prevalence of suicidal ideation, depression, and borderline personality were rather wide; we determined the prevalence of 30% for calculating the sample size. The minimum number of the sample size was 323, required to complete the online survey to ensure a power of 80% and a 0.05 type I error. The number of responses totaled 355. Among them, 13 were excluded: 5 respondents were older than the inclusion criteria (25 years old) and 8 comprised repeat responses. The final number of participants was 342. Of the 342 participants, 336 (98.25%) completed the questionnaires and their data were used for analysis.

Instruments

In addition to the sociodemographic data, e.g., age, sex, number of years of studying, income, etc., the participants were asked to complete the following measurements.

Screening Instrument for Borderline Personality Disorder (SI-Bord)

The SI-Bord consists of 5 questions addressing the salient criteria of DSM-5 of BPD for university students but was modified based on the Thai cultural context [48]. It includes (i) abandonment avoidance; (ii) interpersonal relationships instability; (iii) identity disturbance; (iv) suicidal and self-harm behaviors; and (v) affective instability. The SI-Bord uses a 4-point Likert scale, ranging from never (0) to very often (3). Samples of the questionnaires include “My feelings suddenly changed, such as ‘I don’t know who I am,’ ‘I don’t know where I am going’, ‘I feel lonely deep down’ or ‘I have no goal in life’”. The total score ranges from 0 to 15 and higher scores represent more BPD symptoms (BPS). The study sample yielded a Cronbach’s alpha of 0.76.

2.Revised Thai Multidimensional Scales of Perceived Social Support (r-MSPSS)

This instrument measures the extent to which an individual has experienced being supported by family members, friends, and special individuals [49]. It includes 12 questions using a 7-point Likert scale ranging from very strongly disagree (0) to very strongly agree (6). The higher the score, the higher the level of perceived social support is attained. The revised Thai version demonstrates good psychometric properties [50]. The study sample yielded a Cronbach’s alpha of 0.91.

3.Thai Version of Perceived Stress Scales (T-PSS-10)

This instrument measures the extent to which an individual perceived stress over the past four weeks. It comprises 10 questions with a 5-point Likert scale ranging from never (0) to very often (4). The higher the total score, the higher the level of feeling stress is attained. The T-PSS-10 demonstrates good psychometric properties [51]. The study sample yielded a Cronbach’s alpha of 0.85.

4.Patient-Health Questionaire-8 (PHQ-8)

This instrument measures the extent to which an individual has experienced depressive symptoms over the past two weeks [52,53]. The PHQ-8 differs from the more commonly used PHQ-9 in that it removes the last question addressing suicidal ideation. It contains 8 questions with a 4-point Likert scale ranging from 0 (not at all) to 3 (nearly every day). The higher the total score, the higher the level of depressive symptoms is attained. The study sample yielded a Cronbach’s alpha of 0.89.

5.Suicidality Ideation

The item of suicidal ideation was drawn from the Patient-Health Questionaire-9 (PHQ-9). This item asks the respondent to rate how much s/he feels about “Thoughts that you would be better off dead, or thoughts of hurting yourself in some way?” The 4-response Likert scale ranges from response options 0 (not at all) to 3 (nearly every day). Because it comprises a single item, no internal consistency was provided.

Statistical Analysis

To test the hypothesis, we used a regression analysis. As the outcome, suicidal ideation, was ordinal, ordinal regression or an ordinal logistic regression analysis was applied instead of binary logistic regression to maintain information about the ordering.

Ordinal regression is used to predict an ordinal dependent variable given one or more independent variables. We used ordinal regression to predict the suicidal ideation, measured on a 4-point Likert item from “not at all” to “nearly every day”, based on four independent variables: depressive symptoms (PHQ-8), borderline personality symptoms (SI-Bord), and perceived stress (PSS), and perceived social support (r-MSPSS). In this case, we used ordinal regression to determine whether these independent variables predict the ordinal dependent variable, “suicidal ideation”, where suicidal ideation was determined by categorizing the suicide items in three levels; 0 = no suicidal ideation, 1 = mild, and 2 = moderate to severe (calculated by combining scores 2 and 3). As the independent variables were continuous, we interpreted how a single unit increase or decrease in that variable (e.g., a score increase or decrease in the SI-Bord score), was associated with the odds of suicidal ideation having a higher or lower value (e.g., a one score increase in participants’ SI-Bord score increases the odds that they would consider suicidal ideation to be at a higher level).

Before we began looking at the effects of each explanatory (independent) variable in the model, we had to consider whether the model improved the ability to predict the outcome. To do that, we compared the baseline or ‘Intercept Only’ model (a model without any explanatory variables) against the ‘Final’ model (the model with all the explanatory variables) to see whether it had significantly improved the fit to the data.

The model fitting information containing the −2 Log likelihood for an intercept only (or null) model and the final model (containing the full set of predictors) was analyzed. A likelihood ratio chi-square test was used to test whether significant improvement was indicated in the fit of the final model relative to the intercept only model.

“Goodness of Fit” was illustrated to determine whether a model exhibited good fit to the data. Nonsignificant test results were indicators that the model fit the data well (Field, 2018; Petrucci, 2009). Pseudo-R-square values, unlike the R-square value in the ordinary least square regression, were used in the model fit. McFadden pseudo-R-square values 0.2–0.4 were considered a good model fit [54]. The regression coefficients were used for the predicted change in log odds of being in a higher (as opposed to a lower) group/category on the suicidal ideation level (controlling for the remaining independent variables) per unit increase on the predictors. Odds ratios were also calculated based on regression coefficients, reflecting the multiplicative change in the odds of being in a higher level of suicidal ideation for every unit increase on each particular predictor, holding the remaining predictors constant. An odds ratio > 1 suggested an increasing probability of being in a higher level concerning suicidal ideation as values on a predicting variable increased, whereas a ratio < 1 suggested a decreasing probability with increasing values regarding a predicting variable. An odds ratio = 1 suggested no predicted change in the likelihood of being in a higher category as values on a predicting variable increased.

For sociodemographic data such as sex, year of education, and total scores of psychological variables such as SI-bord score and PSS score, descriptive statistics, i.e., frequency, percentage, mean, and SD were used. A significance level at *p* < 0.05 was considered acceptable. All analyses were performed using IBM SPSS (manufacturer, city and country), Version 22. MedCalc (manufacturer, city and country), Version 19.7 (MedCalc Software, Mariakerke, Belgium) was used to create the graphs.

## 3. Results

Of 336 participants, 275 were females (80.4%) with a mean age of 20.25. (SD = 1.4). The majority of participants comprised health science students. Among all, 14.3% had suicidal ideation. The median score of suicidal ideation was 0, and the interquartile range was 0. Other characteristics are shown in Table 1.

Table 2 shows the difference among variables based on the level of suicidal ideation. Sociodemographic variables included the number of years studying, and the higher risk for suicidal ideation (*p* < 0.05). Non-health science students reported higher levels of suicidal ideation than health science students (*p* < 0.001). For psychological variables, the scores of the SI-Bord, TPSS, and PHQ-8 were higher in higher levels of suicidal ideation, and vice versa for the r-MSPSS (all *p* < 0.001) (Figure 1).

We further conducted a univariate regression analysis for each variable to determine what variables should be included in the multiple ordinal regression model. Finally, we selected age, number of years studying, academic major, as well as the total scores of SI-Bord, r-MSPSS, TPSS, and PHQ-8.

Table 3 shows the univariable regression analysis results. All models fit the data well indicated by the nonsignificance of the Pearson chi-square test and the deviance test results. The standardized regression coefficients ranged from 0.086 to 1.299. All except r-MSPSS exhibited positive associations with suicidal ideation.

For the multivariable regression analysis as shown in Table 4, the model fitting information using a likelihood ratio chi-square test revealed a significantly improved fit of the final model relative to the intercept only (null) model (χ^2^ (6) = 127.66, *p* < 0.001). Then the “Goodness of Fit” was confirmed by the nonsignificance of the Pearson chi-square test (χ^2^ (663) = 409.82, *p* = 1.000) and the deviance test (χ^2^ (664) = 207.57, *p* = 1.000). Pseudo-R-square values were as follows: Cox and Snell = 0.316, Nagelkerke = 0.501, McFadden = 0.381, also indicating that the model displayed a good fit.

The regression coefficients were interpreted as the predicted change in log odds of being in a higher category concerning the suicidal ideation variable (controlling for the remaining predicting variables) per unit increase on the predicting variables. All, except r-MSPSS, were significant positive predictors of the presence of suicidal ideation. PHQ-8 demonstrated a coefficient of 0.149, denoting a predicted increase of 0.149 in the log odds of a student being in a higher category concerning suicidal ideation. In other words, an increase in depressive symptoms was associated with an increase in the odds of suicidal ideation, with an odds ratio of 1.16 (95% CI, 1.05 to 1.22), Wald χ^2^ (1) = 7.80, *p* < 0.01. The same was true for TPSS (Wald χ^2^ (1) = 5.297, *p* < 0.05), SI-Bord (Wald χ^2^ (1) = 4.476, *p* < 0.05), and r-MSPSS scores (Wald χ^2^ (1) = 4.575, *p* < 0.05). For r-MSPSS, an increase in r-MSPSS scores was associated with a decrease in the odds of suicidal ideation, with an odds ratio of 0.97 (95% CI, 0.94 to 1.00).

Among all predictors, SI-Bord scores showed the highest effect size. Age, number of years of studying, and academic major became nonsignificant predictors in the model.

## 4. Discussion

This study aimed to examine the relevant psychosocial variables as predictors for suicidal ideation among these young adults. The findings support related studies, particularly, the role of BPS feelings of stress, and of support concerning suicidal ideation [36]. This sheds some light on the role of BPS on suicidal ideation, in that both depressive symptoms and BPS were highly, significantly associated with suicidal ideation. Consistent with a related study, BPS appeared to have a stronger effect size than depressive symptoms [55]. In other words, even without depressive symptoms, suicidal ideation remained due to BPS. This was conceivably understood because borderline personality usually involves low self-esteem and suicidality. Some characteristics reverberating intra- and interpersonal difficulties, among them feelings of emptiness and interpersonal relationship, can lead to suicidal ideation or even attempts.

Some may argue that suicidality is one part of depression. This is absolutely true as it could be found in high levels of depression based on symptom hierarchy [56]. However, for those exhibiting borderline personality, suicidality may be present in a milder form of depression [57,58]. Many studies did not exclude the suicidal item from the whole depressive questionnaire, rendering a falsely inflated correlation between depressive symptoms and suicidal ideation. That is why we used the PHQ-8, a depression measurement without suicidality, in this study to ensure that the duplication bias was removed and allowing us to observe the true effect size of depressive symptoms on suicidal ideation. Taking all variables together, our findings conveyed an important message that suicidal ideation among university students should be screened for BPS, exclusively from depressive symptoms.

In line with related studies, younger age was associated more with suicidal thought [59], which could have been contributed to the fact that these students had less experience and needed more time for adjustments. The fact that the higher level of studying meant a higher probability to have suicidal ideation may have contributed to an exposure to more stress, especially among health science students [60,61]. Some investigators have suggested that higher education levels may be related to burnout and depression, leading to suicidal ideation, especially among medical and nursing students [1,62,63,64]. However, in this study, age, years of study, and the study major were overshadowed by psychosocial variables. Moreover, none of these mentioned studies included BPS in their studies, and some investigators assumed that suicidal ideation was only one part of depression.

Notably, perceived social support, the only positive psychosocial variable, was shown to provide a protective effect. Growing evidence supports that perceived social support exists among university students with mental health problems. During the COVID-19 pandemic, a high prevalence of mental health issues was observed among students undergoing quarantine. A study with a large sample size revealed that the prevalence of suicidal thoughts was 11.4%, a high level of perceived stress was 4.7%, and severe depression was 16.1% [65]. Another study among university students conducted in three Asian countries including Thailand [66] found a correlation between perceived support and suicidal thoughts among Indonesians and Thais, but not Taiwanese [66]. However, a recent study excluded BPS from their investigation [66]. We assumed, especially in the latter study, that BPS would come into play when different culture/countries become an issue as the personality factor varies from country to country [67].

Our study demonstrated that the model comprising important factors related specifically to university students, especially BPS, appears to be omnipresent in the present day. Suicidal ideation is a small and known part of the larger part, depression, evoked by the imbalance between stressor and social support, whereas most underlying parts might comprise borderline personality. To tackle suicidality, these important variables should be identified in a survey or screening process. Further, providing social support and reducing stress are essential methods to reduce depression and suicidal ideation. Strategic plans should be implemented to help mitigate BPS as well as to promote positive strengths, especially during the early years of university life. Dealing with personality issues would present a tangible challenge, but to protect the student’s wellbeing, this mission may be unavoidable. Overall, our research emphasizes and highlights not overlooking borderline personality disorder in these populations and would like to create the clinicians’ or healthcare providers’ awareness to identify such problems before any intervention. In fact, there have already been many interventions as well as numerous screening tools for borderline personality. What our research adds here is to remind and demonstrate the importance of borderline personality among these young students.

Strengths and Limitations

This study is, to the best of our knowledge, one of the early research reports exploring the relationship between suicidal ideation and variables specific to young adult university students, i.e., BPD symptoms, perceived stress, depressive symptoms, and perceived social support.

This study encountered several limitations. First, the single item suicidality measurement may pose some risk in reliability. A more reliable multiple item measurement for evaluating suicidality is encouraged for further studies. Second, we used the overall, but not specific, type of perceived social support, i.e., family members, friends, and special individuals. As this does not render an opportunity to examine the influence of each type of social support on this age group, a separate analysis should be considered in further studies. Third, social desirability bias may be inevitably present in such self-reporting.

## 5. Conclusions

These findings of suicidal ideation among university students underscore the need for covering the relevant psychosocial factors. Although depression is a major contributor to suicidal ideation, our findings showed that borderline personality symptoms could at least equally be held accountable. The associated variables, i.e., high level of perceived stress, and poor social support were also potent predictors and should not be overlooked. They should be included in further interventions and prevention plans regarding suicidality among university students.

## Figures and Tables

**Figure 1 healthcare-09-01399-f001:**
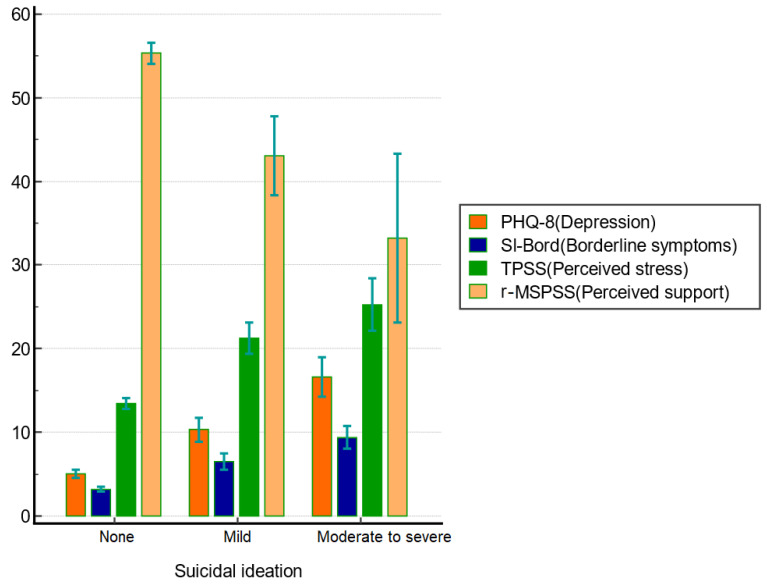
Clustered bar graphs illustrating mean and confidence interval for each group using a bar chart with error bars. r-MSPSS = Revised Thai Multidimensional Scale of Perceived Social Support, PHQ-8 = Patient-Health Questionaire-8, SI-Bord = Short Instrument for Borderline Personality Disorder, T-PSS-10 Thai Version of Perceived Stress Scales.

**Table 1 healthcare-09-01399-t001:** Characteristics and demographic data of the overall participants (N = 336).

Variable		Mean (SD) or n (%)
Age		20.26 ± 1.4
Sex	Female	270 (80.4)
Number of years studying		2.58 ± 1.3
Academic major		
	Health Science	254 (75.6)
	Non-health Science	82 (24.4)
Monthly allowance (THB)		
	<5000	182 (54.2)
	5001–10,000	123 (36.6)
	>10,000	31 (9.2)
Satisfaction with monthly allowance		
	Yes	258 (76.8)
	No	78 (23.2)
Psychological variable		
	Level of suicidal ideation	
	None	288 (85.71)
	Mild	33 (9.82)
	Moderate and severe	15 (4.46)
	SI-Bord	3.80 ± 2.9
	r-MSPSS	53.09 ± 12.8
	TPSS	14.74 ± 6.5
	PHQ-8	6.27 ± 4.9

r-MSPSS = Revised Thai Multidimensional Scale of Perceived Social Support, PHQ-8 = Patient-Health Questionaire-8, SI-Bord = Short Instrument for Borderline Personality Disorder, T-PSS-10 Thai Version of Perceived Stress Scales.

**Table 2 healthcare-09-01399-t002:** Characteristics and demographic data of the participants according to the level of suicidal ideation.

		Level of Suicidal Ideation			Test Difference
Variable		None	Mild	Moderate/Severe	
		N = 288	N = 33	N = 15	
Age		20.20 ± 1.3	20.42 ± 1.7	21.07 ± 1.4	*F* (2, 333) = 3.06, *p* = 0.048
Sex	Female	236 (87.4)	25 (9.3)	9 (3.3)	χ^2^ = 4.84, df2, *p* = 0.089
Number of years studying		2.51 ± 1.2	2.85 ± 1.4	3.4 ± 1.2	*F* (2, 333) = 4.45, *p* = *0*.012
Academic major					
	Health Science	229 (90.2)	19 (7.5)	6 (2.4)	χ^2^ = 18.51, df2, *p* < 0.001
	Non-health Science	59 (72.0)	14 (17.1)	9 (11.0)	
Monthly Allowance (THB)					
	<5000	163 (89.6)	13 (7.1)	6 (3.3)	χ^2^ = 6.65, df4, *p* = 0.155
	5001–10,000	102 (82.9)	14 (11.4)	7 (5.7)	
	>10,000	23 (74.2)	6 (19.4)	2 (6.5)	
Satisfaction with monthly allowance					
	Yes	227 (88.0)	21 (8.1)	10 (3.9)	χ^2^ = 4.73, df2, *p =* 0.094
	No	61 (78.2)	12 (15.4)	5 (6.4)	
Psychological variable					
	SI-Bord	3.19 ± 2.5	6.52 ± 2.8	9.40 ± 2.5	*F* (2, 333) = 64.06, *p* < 0.001
	r-MSPSS	55.35 ± 10.9	43.06 ± 13.2	33.20 ± 18.2	*F* (2, 333) = 40.09, *p* < 0.001
	TPSS	13.42 ± 5.8	21.24 ± 5.3	25.27 ± 5.6	*F* (2, 333) = 54.51, *p* < 0.001
	PHQ-8	5.01 ± 4.1	10.33 ± 4.0	16.6 ± 4.2	*F* (2, 333) = 74.46, *p* < 0.001

r-MSPSS = Revised Thai Multidimensional Scale of Perceived Social Support, PHQ-8 = Patient-Health Questionaire-8, SI-Bord = Short Instrument for Borderline Personality Disorder, T-PSS-10 Thai Version of Perceived Stress Scales.

**Table 3 healthcare-09-01399-t003:** Results of univariable ordinal regression analysis.

	Estimate	S.E.	Wald	df	*p*-Value	95% Confidence Interval	
						Lower Bound	Upper Bound
Age	0.224	0.111	4.041	1	0.044	0.006	0.442
Year	0.319	0.120	7.035	1	0.008	0.083	0.555
Health Science	1.299	0.321	16.337	1	0.000	0.669	1.929
PHQ-8	0.332	0.040	69.018	1	0.000	0.254	0.410
TPSS	0.276	0.035	60.647	1	0.000	0.207	0.346
SI-Bord	0.482	0.059	65.733	1	0.000	0.365	0.598
r-MSPSS	−0.086	0.012	49.698	1	0.000	−0.110	−0.062

S.E. = Standard Error, r-MSPSS = Revised Thai Multidimensional Scale of Perceived Social Support, PHQ-8 = Patient-Health Questionaire-8, SI-Bord = Short Instrument for Borderline Personality Disorder, T-PSS-10 Thai Version of Perceived Stress Scales.

**Table 4 healthcare-09-01399-t004:** Results of multivariable ordinal regression analysis.

	Estimate	S.E.	Wald	df	*p*-Value	95% Confidence Interval		
						Lower Bound	Upper Bound	Odds Ratio (95% CI)
Age	−0.074	0.251	0.087	1	0.768	−0.567	0.419	0.93 (0.59–1.46)
Year	0.130	0.279	0.218	1	0.640	−0.417	0.677	1.14 (0.67–1.93)
Health Science	0.700	0.396	3.115	1	0.078	−0.077	1.476	2.01 (0.93–4.36)
PHQ-8	0.149	0.053	7.800	1	0.005	0.044	0.253	1.16 (1.05–1.22)
TPSS	0.104	0.045	5.297	1	0.021	0.015	0.193	1.11 (1.01–1.22)
SI-Bord	0.170	0.080	4.476	1	0.034	0.013	0.328	1.19 (1.01–1.40)
r-MSPSS	−0.033	0.015	4.575	1	0.032	−0.062	−0.003	0.97 (0.94–1.00)

S.E. = Standard Error, C I = Confidence Interval, r-MSPSS = Revised Thai Multidimensional Scale of Perceived Social Support, PHQ-8 = Patient-Health Questionaire-8, SI-Bord = Short Instrument for Borderline Personality Disorder, T-PSS-10 Thai Version of Perceived Stress Scales.

## Data Availability

The datasets used and/or analyzed during the current study are available from the corresponding author upon reasonable request.

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
