# Peer review of "Borderline Personality Symptoms: What Not to Be Overlooked When Approaching Suicidal Ideation among University Students"

_healthcare, 2021, doi:10.3390/healthcare9101399_

Round 1
Reviewer 1 Report
The research is ambitious in that it builds on previous research to examine factors associated with borderline personality ideation through a survey of university e.g., undergraduates using a very limited set of respondent characteristics and five scales. The range of respondent characteristics is limited and is” representative” of the population and have metrics is not clear why they were chosen, e.g., academic major, monthly allowance. With respect to the four independent scales justification for inclusion should be specifically referenced in the text.
The survey design is cross-sectional. More information is required in order that the reader has a complete understanding of the procedures and methods. The “N” is 326 undergraduates presumably queried in 2019 (Ln 92). What is the universe/population which the sample represents? What metrics, if any, were calculated to conclude representativeness? What is the completion rate? Why does academic major include only two categories and why is there such many health science majors? Exclusion criteria (Ln 104) are cited; the authors need to explain exactly how respondents were excluded – in advance with some prior knowledge or through the survey instrument? What number and percent were excluded based on some measure of population or surveys received.
Table 1 is difficult to read and should be reformatted along with other changes. Specifically, descriptive statistics includes all variables including suicidal ideation using only the “Overall” column. Additionally, and unfortunately the ordinal properties of each scale are obscured using summary (mean) statistics. I would like to have more information of the distribution of scores for the scales and commentary in the text. Next, create a new Table 2 using the “Suicidal Ideation” and “Test Differences” columns from Table 1.
The analysis is straightforward and the results. Do the four psychological scales exhibit any pattern of correlation?
There are several minor issues. When BPD is used for the first time it should be preceded by “borderline personality disorder;” likewise for BPS. Line 82 sentence does not have a citation. In Line 192 “faculty attended” is not clear as to what it means nor of its origin. The phrase (Ln 264) “tip of the iceberg is colloquial. Line 233 requires citation. I suggest a thorough edit.
Author Response
Reviewer 1
The research is ambitious in that it builds on previous research to examine factors associated with borderline personality ideation through a survey of university e.g., undergraduates using a very limited set of respondent characteristics and five scales. The range of respondent characteristics is limited and is” representative” of the population and have metrics is not clear why they were chosen, e.g., academic major, monthly allowance. With respect to the four independent scales justification for inclusion should be specifically referenced in the text.
Response We have admitted that there was limitation regarding the outcome used for the study, which we have mentioned in the limitation section. Despite that, we thought it is still worth reporting as the results are important and urgently need to get awareness from people involved with helping the students. Some variables are significant based on the literature review such as the academic major (e.g., medical students and nurses are prone to have more problems). Another issue important to students especially low-middle income countries is monthly allowance. Research shows that the perception of sufficiency of income (not the exact amount of money), plays a more important role on their mental health wellbeing (e.g., psychological distress among medical students in conflicts: a cross-sectional study from Syria - PubMed (nih.gov)). This is why we have added these variables in the study.
1)The survey design is cross-sectional. More information is required in order that the reader has a complete understanding of the procedures and methods. The “N” is 326 undergraduates presumably queried in 2019 (Ln 92). What is the universe/population which the sample represents? What metrics, if any, were calculated to conclude representativeness? What is the completion rate? Why does academic major include only two categories and why is there such many health science majors? Exclusion criteria (Ln 104) are cited; the authors need to explain exactly how respondents were excluded – in advance with some prior knowledge or through the survey instrument? What number and percent were excluded based on some measure of population or surveys received.
Response We have revised this part as follows
- Design
This study employed a cross-sectional online survey of 336 undergraduate students in Thailand in late 2019 before the COVID-19 pandemic in Thailand. It comprised a cross-sectional online survey. The study was approved by the Ethics Committee of the Faculty of Medicine at Chiang Mai University, Thailand.
Participants
Undergraduate students throughout Thailand were invited to participate in the study. The inclusion criteria included age 18–25 years, studying at undergraduate level, fluent in writing and speaking Thai, and able to access the Internet. The exclusion criteria included being diagnosed with schizophrenia, bipolar disorder, drug or alcohol use disorder, and being intoxicated with alcohol within 24 hours before participating in the study. After finalizing a written informed consent form, the participants were asked to complete questionnaires on the Internet via personal computer, laptop, smartphone, or tablet. The amount of 100 THB (3 USD) was given to each participant for compensation after they had submitted their answers.
The target sample size of participants was estimated based on related studies among university students. As reviewed, the range of the prevalence of suicidal ideation, depression, and borderline personality were rather wide; we determined the prevalence of 30% for calculating the sample size. The sample size required to complete the online survey to ensure a power of 80% and a 0.05 type I error, therefore the minimum number was 323. The number of responses totaled 355. Among them, 13 were excluded: 5 respondents were older than the inclusion criteria (25 years old) and 8 comprised repeat responses. The final number of participants was 342. Of 342 participants, 336 (98.25%) completed the questionnaires and their data were used for analysis.
- b) The majority of participants were health science students (6%), the rest were technology (7.9%), social science (8.2%), and others (8.3%). We have added this data in the manuscript. The reason the most are health science student is interesting. We are uncertain about such results, but we thought that it could be because it was an online survey related to mental health, it might have caught the attention to health science students more than nonhealth science students.
2) Table 1 is difficult to read and should be reformatted along with other changes. Specifically, descriptive statistics includes all variables including suicidal ideation using only the “Overall” column. Additionally, and unfortunately the ordinal properties of each scale are obscured using summary (mean) statistics. I would like to have more information of the distribution of scores for the scales and commentary in the text. Next, create a new Table 2 using the “Suicidal Ideation” and “Test Differences” columns from Table 1.
Response Thank you for your suggestion we have created a new Table 2 suggested.
2) The analysis is straightforward and the results. Do the four psychological scales exhibit any pattern of correlation?
Response Yes, they are, but we did not show the correlation among them as our focus was to prove how each psychological variable predicted the suicidal ideation.
3) There are several minor issues. When BPD is used for the first time it should be preceded by “borderline personality disorder;” likewise for BPS. Line 82 sentence does not have a citation. In Line 192 “faculty attended” is not clear as to what it means nor of its origin. The phrase (Ln 264) “tip of the iceberg is colloquial. Line 233 requires citation. I suggest a thorough edit.
Response. We apologize for the mistakes and thank you for this suggestion. We have corrected BPD and BPS that are mentioned for the first time. Add citations on line 82 (now in line 94) and 233 (now in line 281). The word “faculty attended” has been changed to “academic major”. “Tip of the iceberg” in the text has been revised as follows “Suicidal ideation is a small and known part of the larger part, depression, evoked by the imbalance between stressor and social support, whereas most underlying parts might comprise borderline personality.”
Finally, we had our manuscript edited again by an English native speaker.

Reviewer 2 Report
This is a very well-reasoned and well written paper on a topic of major importance. The research design is sound and the data analyses seem to be appropriate -- I say "seem to be" because I am not well versed on some of the statistics. The measures used are appropriate and the conclusions are solidly in line with the findings reported. The authors are aware of the study's limitations, most notably the single item suicidal ideation measure. Despite the limitation, the importance of this research warrants publication.
As noted above, I feel the study that is reported is a fine piece of work that most definitely should be published. The suggestion that I note here focuses primarily on what the authors might consider for a follow-up study. In subsequent research it would be useful to focus more on thoughts, feelings, schemas that underlie the more symptomatic measures used in this research. I have in mind the kind of work that was reported in the 2020 study of beliefs underlying schizotypal personality disorder. It would also be interesting to further investigate mediation models of suicidal ideation as was reported in reference #26 and other recent studies on how beliefs mediate the relationship between childhood trauma and subsequent psychopathology. These are simply suggestions for future work and in no way detract from the clinical and health policy significance of the current paper.
Author Response
Reviewer 2
This is a very well-reasoned and well written paper on a topic of major importance. The research design is sound, and the data analyses seem to be appropriate -- I say "seem to be" because I am not well versed on some of the statistics. The measures used are appropriate and the conclusions are solidly in line with the findings reported. The authors are aware of the study's limitations, most notably the single item suicidal ideation measure. Despite the limitation, the importance of this research warrants publication.
As noted above, I feel the study that is reported is a fine piece of work that most definitely should be published. The suggestion that I note here focuses primarily on what the authors might consider for a follow-up study. In subsequent research it would be useful to focus more on thoughts, feelings, schemas that underlie the more symptomatic measures used in this research. I have in mind the kind of work that was reported in the 2020 study of beliefs underlying schizotypal personality disorder. It would also be interesting to further investigate mediation models of suicidal ideation as was reported in reference #26 and other recent studies on how beliefs mediate the relationship between childhood trauma and subsequent psychopathology. These are simply suggestions for future work and in no way detract from the clinical and health policy significance of the current paper.
Response: Thank you for your encouragement and suggestion. We definitely agree with your idea especially the beliefs underlying borderline personality symptom, which may vary among individuals presenting the same symptoms. Also, we highly appreciate your suggestion for the further study regarding how beliefs mediate the relationship between childhood trauma and subsequent psychopathology. I believe that would be of great value for both clinical and research.

Reviewer 3 Report
While I commend the authors for aiming to address suicidal depression, this manuscript fails to detail the necessary scientific methods used. There is no mention of the approach for solving the problem or even the path to analyzing the data. This paper seems more of a course project report. The authors are sincerely recommended to expand the paper to discuss the methods for addressing depression, motivate the novelty in their approach and the need for real-user evaluation. The goals for experimentation work and evaluation have to be discussed at length.
Author Response
Reviewer 3
While I commend the authors for aiming to address suicidal depression, this manuscript fails to detail the necessary scientific methods used. There is no mention of the approach for solving the problem or even the path to analyzing the data. This paper seems more of a course project report. The authors are sincerely recommended to expand the paper to discuss the methods for addressing depression, motivate the novelty in their approach and the need for real-user evaluation. The goals for experimentation work and evaluation have to be discussed at length.
Response. We thank for your concern. We believe that we have covered all the six steps of the scientific method including 1) asking a question 2) conducting background research 3) constructing a hypothesis, 4) experimenting to test the hypothesis, 5) analyzing the data from the experiment and drawing conclusions, and 6) communicating the results.
The reviewer has mentioned our research is a descriptive and correlational study rather than experimental study. We have revised the reason to use such method of analysis to make it clearer as follows.
“To test the hypothesis, we used regression analysis. As the outcome, suicidal ideation, was ordinal, ordinal regression or ordinal logistic regression analysis was applied instead of binary logistic regression to maintain information about the ordering”
Regarding the comment “the goals for experimentation work and evaluation have to be discussed at length”, if we understood correctly, we believe that this research is more like confirming research rather than offering a novelty of approach. However, we have added this part as suggested to the discussion
Overall, our research aims to emphasize and highlight the overlooking borderline personality disorder in these populations and would like to draw the clinicians’ or healthcare providers’ attention to identifying such problems before any intervention. In fact, there has already been a lot of interventions as well as many screening tools for borderline personality. What our research adds here is to remind and show the importance of borderline personality among these young students.

Round 2
Reviewer 1 Report
Thank you for responding to my previous concerns. The explanations are satisfactory.
Author Response
Thank you
Reviewer 3 Report
The authors have fairly addressed the reviewers' comments. The paper in its current form reads much better. However, few more improvements can be made:
- The regression analysis conducted needs to be explained in more detail.
- The distribution across the dataset can be represented visually than a table so as to observe key differences.
- If most of the study participants were female, how does that impact the nature of the study and the inferences from the results? What about selection bias?
- The results are presented in a holistic fashion instead of breaking it down into parts which can help understand different characteristics within the datasets.
Author Response
Dear Editor and Reviewers
We thank the editor and reviewers for their useful comments and suggestion. Please see below our point-by-point response to those comments.
Reviewer 3
The authors have fairly addressed the reviewers' comments. The paper in its current form reads much better. However, few more improvements can be made:
- The regression analysis conducted needs to be explained in more detail.
Response: We have added the following statements to explain more about the analysis.
Ordinal regression is used to predict an ordinal dependent variable given one or more independent variables. We used ordinal regression to predict the suicidal ideation, measured on a 4-point Likert item from "not at all" to "nearly every day"), based on four independent variables: depressive symptoms (PHQ-8), borderline personality symptoms (SI-Bord), and perceived stress (PSS), and perceived social support (rMSPSS). In this case, we used ordinal regression to determine whether these independent variables predict the ordinal dependent variable, "suicidal ideation", where suicidal ideation was determined by categorizing the suicide items in 3 levels; 0 no suicidal ideation, 1 = mild, and 2 = moderate to severe (calculated by combining scores 2 and 3). As the independent variables were continuous, we interpreted how a single unit increase or decrease in that variable (e.g., a score increase or decrease in SI-Bord score), was associated with the odds of suicidal ideation having a higher or lower value (e.g., a one score increase in participants' SI-Bord score increases the odds that they would consider suicidal ideation to be at a higher level).
Before we began looking at the effects of each explanatory (independent) variable in the model, we had to consider whether the model improved the ability to predict the outcome. To do that, comparing the baseline or ‘Intercept Only’ model (a model without any explanatory variables) against the ‘Final’ model (the model with all the explanatory variables) to see whether it had significantly improved the fit to the data.
In the results section we have revised as follows.
In other words, an increase in depressive symptoms was associated with an increase in the odds of suicidal ideation, with an odds ratio of 1.16 (95% CI, 1.05 to 1.22), Wald χ2(1) = 7.80, p <.01 The same was true for TPSS (Wald χ2(1) = 5.297, p <.05), SI-Bord (Wald χ2(1) = 4.476, p < .05), and rMSPSS scores (Wald χ2(1) = 4.575, p <.05). For rMSPSS, an increase in rMSPSS scores was associated with a decrease in the odds of suicidal ideation, with an odds ratio of 0.97 (95% CI, 0.94 to 1.00).
- The distribution across the dataset can be represented visually than a table so as to observe key differences.
Response: We agree with the usefulness of the visual presentation. We have created Figure 1 comparing the predictors among three groups that we thought would be worthwhile to present.
Figure 1. Clustered bar graphs illustrating mean and confidence interval for each group using a bar chart with error bars
- If most of the study participants were female, how does that impact the nature of the study and the inferences from the results? What about selection bias?
Response: The majority of subjects were female because most university student populations are female. We thought that the proportion in the sample was consistent with a real population.
- The results are presented in a holistic fashion instead of breaking it down into parts which can help understand different characteristics within the datasets.
Response: We appreciate this suggestion. However, we would like to keep it simple rather than break it down into parts. We believe that our analysis is quite simple, that is, testing the regression model for these four interesting predictors, and is already easy to understand.
